# Improved siamese tracking for temporal data association

Yi Tao[1,2], Fei Wang [1]*, Mohan Li[1], Jie Liu[1], Juncheng Zhou[1], Bo Dong[1], Ruidong Liu[1], Sihao Chen[1], Kan Jiao[1]

1 Xi'an Aerospace Automation Co., Ltd., The 6th Academy of China Aerospace Science and Industry Corporation, Xi'an, Shaanxi, China, 2 Xidian University, Xi'an, Shaanxi, China

* wf_asam@163.com

## Abstract

Temporal image data association is essential for visual object tracking tasks. This association task is typically stated as a process of connecting signals from the same object at different times along the time axis. Temporal data association is usually performed before state estimation. The accuracy of data association processing results is fundamental to guaranteeing the correctness of all subsequent procedures. This paper proposes an efficient approach for temporal data association focused on obtaining accurate data association processing results in Siamese network framework. Siamese network has recently achieved strong power in visual object tracking owing to its balanced accuracy and speed. Based on data association processing and multi-tracker collaboration, our algorithm achieves high accuracy and strong robustness, which outperforms several state-of-the-art trackers, including standard Siamese trackers.

## 1. Introduction

Large-scale time series datasets represents a sequence of data points collected or recorded at time intervals and ordered chronologically. Such intricate data collection is imperative for the surveillance of dynamic alterations within multifarious systems, serving as a fundamental element for both analytical review and prospective estimations, thereby delivering significant value in a breadth of fields.

The pertinence of time series datasets is manifold, facilitating profound examinations into enduring tendencies and elucidating the trajectory of specific indices over temporal spans. These data are central to discerning trends, which are particularly salient for entities whose operations are susceptible to temporal variability. Historical data points within these series are central in forecasting imminent events.

Visual tracking [1–13] provides a deep understanding of how objects move and evolve over time, aiding tasks from security surveillance to environmental studies. Visual tracking ascertains the persistent observation of mobile subjects and is particularly advantageous in a range of scenarios, encompassing meteorological surveillance to faunal tracking. This process involves the initial localization of an object and the continuous detection and localization of that object in subsequent frames, even in the presence of complex environmental changes, occlusions, illumination changes, or camera motion.

**Data availability statement:** All relevant data are within the manuscript and its Supporting Information files.

**Funding:** The authors acknowledge financial support for this work from the following sources: Qinchuangyuan High-level Innovative and Entrepreneurial Talent Introduction Project (Grant No. QCYRCXM-2022-183; Principal Investigator: Yi Tao). Shaanxi Provincial Key Research and Development Program Project (Grant No. 2024GX-YBXM-128; Principal Investigator: Yi Tao).

**Competing interests:** The authors have declared that no competing interests exist.

Temporal image data association is an intermediate key step in visual object tracking tasks. In the context of visual object tracking, temporal data can include the location and size of the object. These data points constitute the trajectory of the target in time and space and are essential for understanding and predicting the target behavior. Given that the ultimate outcome of visual object tracking tasks largely hinges on the quality of the association, it is indeed worthwhile to invest in the pursuit of an optimal solution. General, a time-context model trained with information about the target and the surrounding background detects the location of the target.

In video scenes, a multitude of challenges, such as complex environmental changes, occlusions, illumination changes, and camera motion, can significantly impair the continuity of video data as a time series. As illustrated in Fig 1, occlusions, where objects overlap or are partially hidden by other elements within the scene, are particularly problematic. These occlusions can cause tracking algorithms to lose sight of the target, leading to incorrect associations and disrupted temporal coherence, and complicate post-occlusion recovery due to appearance changes or similar objects. Additionally, variations in lighting, weather conditions, and camera motion can introduce noise and distortions, further complicating the tracking process. Fast-moving targets, abrupt changes in direction, and cluttered backgrounds also pose significant obstacles, as they can easily confuse tracking systems and result in lost tracks or false positives. These challenges collectively undermine the reliability and consistency of the time series data, necessitating robust algorithms capable of maintaining accurate and continuous tracking despite these adversities.

Recently, Siamese network has demonstrated strong power in visual tracking owing to its balanced accuracy and speed [14–19]. Siamese network, which is specifically designed to address the image similarity problem, is inherently more appropriate than classical Convolutional Neural Networks (CNNs) for visual object tracking. By formulating the visual tracking task as a matching problem, Siamese tracker, which consists of a template branch and a test branch sharing all parameters in CNN, is trained as a generic similarity function between the two branches on a video dataset. Then Siamese tracker searches for the target in a search region by correlation with a sliding window. The fully convolutional Siamese network (SiamFC) [14] has achieved well tracking performance in the case of no model update at all.

However, relying only on the information of the first frame for visual object tracking restricts the discrimination ability of SiamFC. Naturally, the discrimination capability of SiamFC is weak in case of multiple challenges, e.g., target appearance changes. Temporal image data association is an intermediate key step in visual object tracking tasks. Given that the ultimate outcome of visual object tracking tasks largely hinges on the quality of the association, it is indeed worthwhile to invest in the pursuit of an optimal solution. We propose

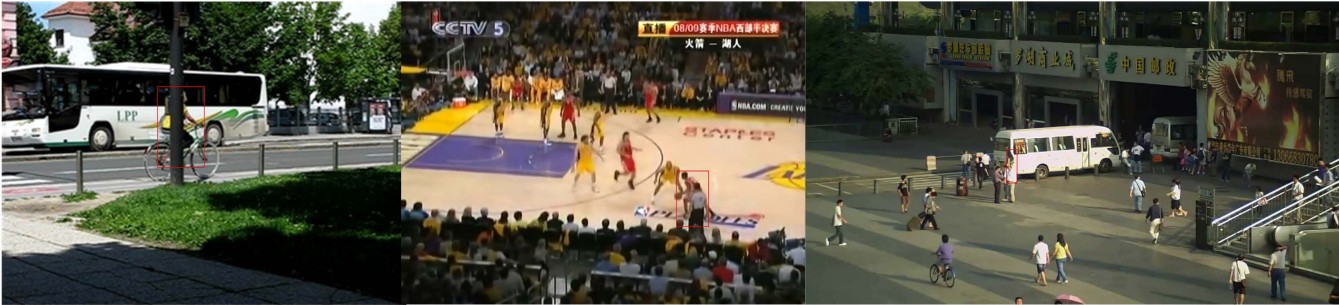

**Fig 1. Occlusion challenge.**

a template ensemble by adapt online reliable update processes to improve tracking performance. We combine multiple templates online trained via different reliable update processes in the Siamese network framework. These templates complement each other. This strategy ensures the template ensemble always containing the previous reliable information, which is effective for improving tracking robustness.

In this paper, we propose a decision-level fusion to combine multiple templates online trained via different reliable update processes into a compound tracker and correct the errors induced by similar distractors. This method ensures the coherence and consistency of time series data, and can construct more accurate motion model to improve the stability and accuracy of tracking. Specifically, we first obtain multiple templates by online training via different reliable update processes in the Siamese network framework and select the appropriate one from them. We implement the reliable update strategy by discriminating the change of the response map. Then we propose to correct the errors from similar distractors to improve tracking robustness. Fig 2 shows that our method effectively improves the performance of our baseline SiamFC. Our collaborative tracking algorithm combines diverse templates online trained via different reliable update processes to achieve an improved tracking performance. And our tracker re-detects the target from multiple target-like regions to maintain the continuity of tracking when the target suffers from background clutter, similar appearance objects, deformations and occlusions. Our contributions are summarized as follows:

- We explore model fusing strategies to make a better use of diverse models via different update processes.

- We harness historical information through a variety of online update mechanisms and corresponding models, each meticulously trained utilizing time-series data, to enhance the precision and robustness of object tracking.

- To maintain the coherence and consistency of time series data, we discriminate the change of the response map to determine whether reliable or not.

- To maintain the continuity of tracking, we re-detect the target from multiple target-like regions to handle background clutter, similar appearance objects, deformations and occlusions.

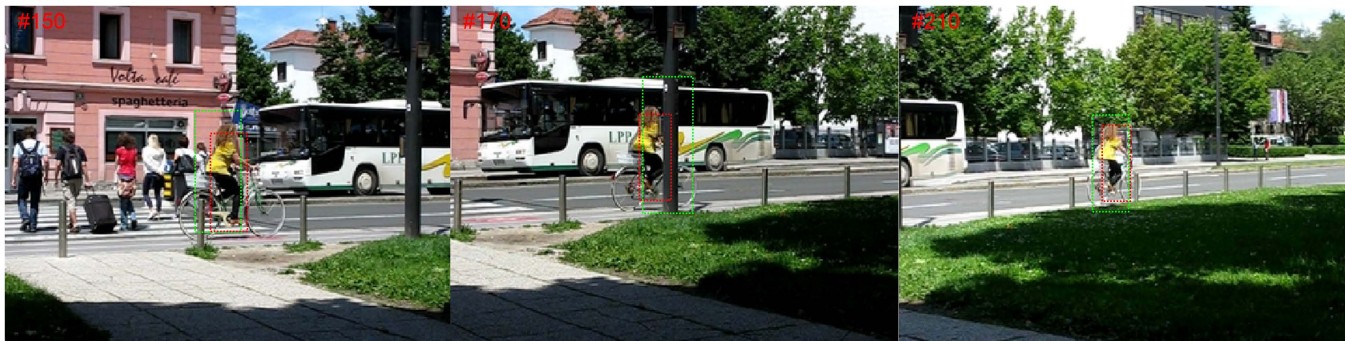

**Fig 2. Performance improvement of SiamFC by the proposed algorithm in this chapter.**

## 2. Related work

### 2.1. Deep similarity tracking

Siamese trackers formulate visual object tracking as a generic similarity problem. First, a model is trained as a generic similarity function between the two branches on a video dataset during an offline phase. Then the model is applied to evaluate the similarity by correlation between two network inputs: the target template and the current frame. SiamFC [14] trains two identical fully convolutional networks to represent object and search area, and further generate the tracking result through finding the maximum value of correlation response map.

SiamFC outperformed several state-of-art trackers, while achieving real-time speed. Several improvements were subsequently proposed. For example, rather than performing correlation on deep features directly, correlation filter network (CFNet) [20] trained a correlation filter based on the extracted features of object to speed-up tracking. Siamese network with semantic and appearance features (SA-Siam) [16] encoded the target by a semantic branch and an appearance branch to improve tracking robustness. But since these Siamese trackers only use the output of the last convolutional layers, more detailed target specific information from earlier layers is not exploited. In our work, we propose a Siamese tracker that combines features from different hierarchical levels.

### 2.2. Online updating with time series data

The fixed models of the original Siamese trackers are more likely to fail in the case of target appearance changes. This weakness can be improved by online update. Some discriminative correlation filter (DCF) based trackers employ a straightforward linear strategy that exhibits simplicity in implementation while ensuring efficiency with regard to memory consumption and computational complexity. This approach effectively updates the object appearance model through the utilization of feature extraction from each frame. This strategy assumes that the object appearance changes in a fixed rate in consecutive frames. Inspired by this strategy, several Siamese trackers [17,21,22] online update the object appearance template with a fixed learning rate across all frames in the video. However, some contaminated templates, which may lead to model drift and even tracking failure, are undesirable under severe occlusions, for example. For overcoming this problem, we propose two conservative strategies to apply the previous reliable information for the object template branch update.

$$\mathfrak{u}\left(T_t\right) = \left(1-\alpha\right)\mathfrak{u}\left(T_{t-1}\right) + -\mathfrak{g}\left(\overline{x}_t\right) \tag{1}$$

Where $\mathfrak{u}\left(T_t\right)$ and $\mathfrak{u}\left(T_{t-1}\right)$ are the updated parameters and $\alpha$ is a learning rate. For template-based tracker, g is the identity function. For correlation filter-based tracker, g is $CF(\bullet)$.

### 2.3. Multi-branch tracking fusion

The diversity of target appearances during tracking shows that a single fixed template cannot be discriminative in all tracking situations with varied challenges. In the pursuit of more precise and reliable object tracking, the concept of multi-branch tracking fusion holds significant relevance. As we've established, the ever-changing nature of target appearances demands adaptable tracking mechanisms. Multi-branch tracking fusion, with its various manifestations like Multi-domain convolutional neural networks (MDNet) [23], Tracking adaptation context-aware auto-encoders (TRACA) [24], Multi-branch Siamese tracking (MBST) [25], and Multi-features Siamese tracker (MFST) [26], attempts to counter the challenges posed by diverse target conditions. MDNet [23] pretrain multiple branches using independent

information belonging to different domains. However, a notable shortcoming is that the independence of the domains might lead to suboptimal integration during actual tracking. TRACA [24] trains multiple auto-encoders according to different contexts and selects the best one by a context-aware network. While this context-based selection mechanism is innovative, it suffers from high computational complexity. MBST [25] selects the optimal branch from multiple branches with diverse feature representations according to their response maps. Although it provides a practical way to adapt to varying target appearances, it heavily depends on the quality and diversity of the predefined feature representations. MFST [26] fuses multiple feature representations which are extracted from different layers of two models to improve the tracking performance. This fusion aims to enhance the tracking performance by combining different levels of semantic and visual information. Nevertheless, the challenge lies in determining the optimal combination weights for the different features. An improper weighting scheme can either overemphasize less relevant features or underutilize crucial ones, leading to subpar tracking accuracy.

## 3. Our proposed tracker

As illustrated in Fig 3, we propose a decision-level fusion, which combines multiple templates online trained via different reliable update processes into a compound tracker, and a re-examination method, which re-detects the target misclassified as background by the basic tracking network, to improve the temporal consistency of time series data.

### 3.1. Siamese baseline

In the recent landscape of visual target tracking, tracking algorithms derived from Siamese networks have become a focal point due to their capacity to balance precision with computational efficiency. In this work, we implement SiamFC [14] as our baseline. SiamFC treats the task of visual target tracking as an issue of learning similarities. The theoretical underpinning of SiamFC involves the deployment of two congruent fully convolutional networks tasked to act as a target exemplar and an investigative branch for abstracting features from the target and its proximate search area. Post feature extraction, these attributes are assimilated using a

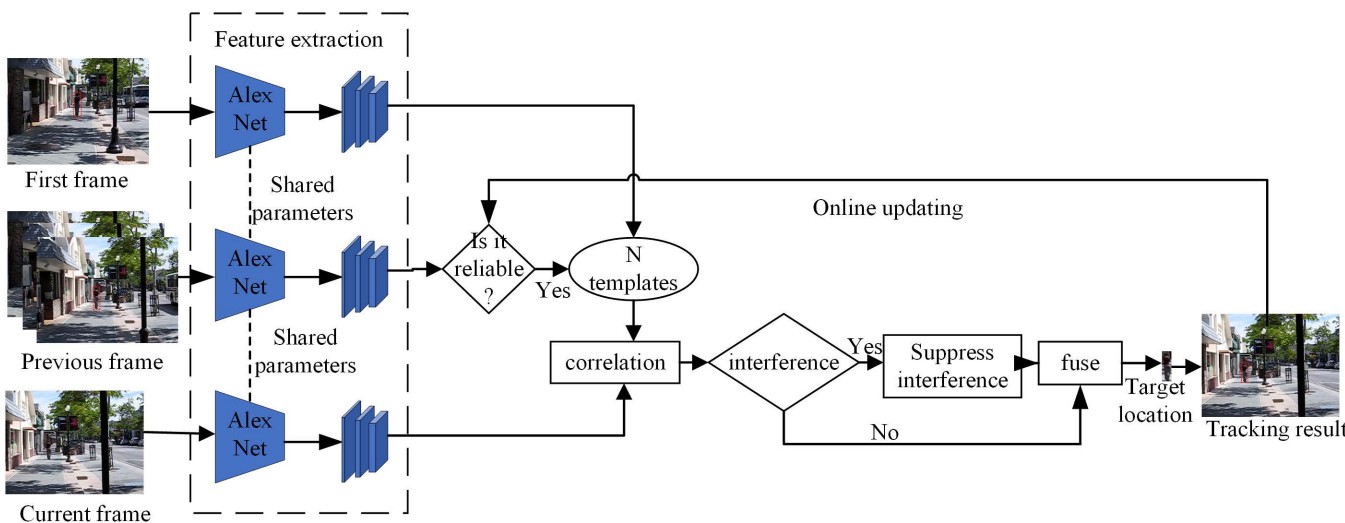

**Fig 3. Overall framework of our proposed tracking algorithm.**

cross-correlation procedure, creating a response map indicative of the correlation. The apex of this map directs to the definitive position of the target for the ensuing tracking activity. Both image patches $x$ and searching areas $z$ are processed by Alex network (AlexNet) [27] $\phi$ and share all parameters to obtain the feature map $\phi_\theta(x)$ and $\phi_\theta(z)$. The response map $f_\theta(x,z)$ is calculated by a cross-correlation function,

$$f_\theta(x,z) = \phi_\theta(x) \cdot \phi_\theta(z) \tag{2}$$

The SiamFC system ascertains the position of the target within the search zone by applying cross-correlation functions between the sample portrayal of the target and a translational scanning through the proposed area, pinpointing the object locus at the highest point on the resultant response map. For the high discriminative power, the network is offline trained by many random pairs in a video dataset, and employs the logistic loss as follows:

$$L\left(f_\theta(x_i,z_i),c_i\right) = \log\left(1 + \exp\left(-c_i g_\theta(x_i,z_i)\right)\right) \tag{3}$$

Where $c_i \in \{+1,-1\}$ is the true label of the image pair, and $g_\theta(x_i,z_i)$ is the actual correlation value between the sample image and the candidate image.

## 3.2. Multi-template trained with time series data fusion

Challenges such as target appearance variations, similar distractors and occlusions, which widely exist in the video dataset, demand more from template branches. The fixed target template extracted from the first frame is not sufficient to handle these challenges. We propose to add online appearance information extracted from the subsequent frames to templates to improve the tracking performance. In the general update strategy, all update possibilities are as shown in Equation 1. Different from this strategy, we initial store all update possibilities and only update the selected template in the current frame.

The target template of the first frame is the only deterministic template that can cope with frequent and severe occlusion challenges. Updating the template complements the fixed template of the first frame and has complementary properties. It is a research problem to efficiently combine online updated templates and fixed templates of the first frame. Reliable sample judgment mechanism is mentioned in the next section to screen reliable samples. Multiple different templates are trained through different updating processes using the screened samples. The template used in the current frame is updated online under the premise of ensuring the reliability of the estimation information of the current frame, while other templates are retained. Fusing these templates gives them the ability to adaptively adjust based on the scene of the current frame, essentially selecting historical sample information for online template training adaptively. The evaluation criterion for multiple template fusion is peak-to-sidelobe ratio (PSR), which is a classic measurement standard used by correlation filtering tracking algorithms to evaluate the discriminative power of models. PSR can accurately select the most discerning template in the template set.

## 3.3. Reliable samples selection policy

Numerous tracking systems incrementally refine their visual models [8,11,20,26] with fresh samples taken from each subsequent frame to accommodate alterations in the observed target. These models consistently adapt at a predetermined learning pace with every new frame, without weighing the dependability of the incorporated samples. This practice can unfortunately lead to model corruption when samples are compromised due to concealment or errors

in tracking. Such corruption may build up progressively and eventually result in significant deviations within the model, or even a total breakdown in tracking capabilities. To effectively filter for high-quality samples, we introduce a method to gauge the trustworthiness of samples from each frame.

Conceptually, a frame that yields dependable samples is characterized by a response map that shows nominal fluctuation. Yet, when the target is obscured or if a misstep in tracking occurs, the response map is likely to exhibit a striking transformation. Therefore, we harness these variations in response map orientation to determine the trust level of each frame. Specifically, the direction of the response map is shown by the rest of 0.5 times the search region excluding 0.4 times the search region around the maximum confidence score. The workable segment is segregated into eight distinct sectors based on their spatial positioning. This generates a series of eight spatial bins arrayed at even intervals throughout the entire 360-degree scope, corresponding to $0–2\pi$ radians. The collective response tallies, viewed as weighted contributions, are grouped into their matching spatial bins relying on their respective positions within the frame.

Then, the total response value $S_k$ of each region is projected to the X-axis and Y-axis to get $D_x$ and $D_y$ according to the angle coefficient corresponding to each region:

$$\begin{cases} D_x = \sum\nolimits_{8}^{k=1} W_k^x S_k \\ D_y = \sum\nolimits_{8}^{k=1} W_k^y S_k \end{cases} \tag{4}$$

Where $W_k^x$ and $W_k^y$ are the corresponding weighting coefficients in the X-axis and Y-axis directions, which are obtained by trigonometric functions according to the approximate angle of the region. The direction angle of the response map $A$ is obtained through the following equation:

$$A = \begin{cases} \tan^{-1} \dfrac{D_y}{D_x}, & y \geq 0 \\ \tan^{-1} \dfrac{D_y}{D_x} + \pi, & y < 0 \end{cases} \tag{5}$$

The difference in direction angles between adjacent frames is inversely mapped to the interval $[1, -3]$ using the following equation:

$$C_A^t = \begin{cases} \cos\left|A^t - A^{t-1}\right|, & \left|A^t - A^{t-1}\right| < \pi \\ -\cos\left|A^t - A^{t-1}\right| - 2, & \left|A^t - A^{t-1}\right| > \pi \end{cases} \tag{6}$$

The change in the direction of the response map between two adjacent frames is described by the following equation:

$$V^t = C_A^t \frac{min\left(\sqrt{\left(D_x^t\right)^2 + \left(D_y^t\right)^2}, \sqrt{\left(D_x^{t-1}\right)^2 + \left(D_y^{t-1}\right)^2}\right)}{max\left(\sqrt{\left(D_x^t\right)^2 + \left(D_y^t\right)^2}, \sqrt{\left(D_x^{t-1}\right)^2 + \left(D_y^{t-1}\right)^2}\right)} \tag{7}$$

## 3.4. Correcting the errors from similar distractors

Under conditions of extreme clutter, frequent occlusions and similar appearance objects, detecting the precise item of interest amidst multiple misleading regions poses a demanding problem. These deceptive elements can often lead to errors in the tracking process, as

differentiating the intended focus from these so-called distractors is not straightforward. Influences from such elements can bias the indicators used to gauge similarities in the tracking templates, potentially leading to a comprehensive breakdown in the tracking methodology. We propose a re-examination method, which re-detects the target misclassified as background by the basic tracking network, and achieves object tracking while filtering out false backgrounds, improving the temporal consistency of the algorithm. We propose to evaluate the confidence of similar target regions using PSR and the local maximum of each peak to determine the final tracking result. Preliminary to this decision process, there is a need to scrutinize the effects caused by peaks that are not in the vicinity of the presumed target, to verify if the distractions have significantly deviated the tracking or if a re-calibration of the target position is justified.The requirement for remediation is triggered under a set of specific conditions: Initially, when the PSR associated with the dominant value in a tailored response visualization—the cosine window—is minimized, reflecting insufficient differentiation capacity by the tracking template. Subsequently, when the difference in magnitude between the primary and secondary peaks on an alternative response map, devoid of the cosine window, is inconsequential, indicating confusion within the template in recognizing the actual focus from the paramount distractor. This level of dissimilarity is captured by a unique quotient, the first-to-second ratio (FSR), representing the relationship between the maximum peak location and the second-peak location without cosine window.Moreover, when there is a significant physical divide between the apex without the cosine feature and the second in command relative to the position with the aforementioned feature, it paints a picture of notable disparity between the mirroring target zone and the true intended area, which if incorrectly identified, could be detrimental to the tracking fidelity.Meeting all these specified conditions signals that the anticipated target has been influenced heavily, necessitating a renewed analysis of its positioning. If, in such a scenario, the PSR of the highest peak on the unmodified response map reigns supreme, it denotes an erroneous initial identification of the target position. Past determinations hinting at a sizable gulf between the falsified position swayed by distractions and the legitimate target space mandates a prompt and vital adjustment, realigning the focus to the position of the peak as charted on the unfiltered response map.

## 4. Experiments

The performance of the proposed tracking algorithm is comprehensively evaluated using three video datasets in this section. Specifically, the datasets used for evaluation are temple color 128 (TC-128) [28], unmanned aerial vehicle 123 (UAV123) [29], and long-term unmanned aerial vehicle 20 (UAV20L) [29], as shown in Table 1. These datasets cover a wide range of scenarios, including pedestrians, close-range toys, and vehicles, ensuring the complexity and diversity of the video material. This variety enables a more comprehensive assessment of the tracking

**Table 1.  Performance of the proposed tracking algorithm and comparison algorithms in three datasets. The higher the metrics, the better.**

|  | Temple-Color | | UAV123 | | UAV20L | |
|---|---|---|---|---|---|---|
|  | DP | AUC | DP | AUC | DP | AUC |
| Proposed | **0.726** | **0.522** | **0.721** | **0.499** | **0.651** | **0.441** |
| SiamFC [14] | 0.668 | 0.490 | 0.696 | 0.477 | 0.606 | 0.399 |
| SRDCF [31] | 0.662 | 0.485 | 0.676 | 0.464 | 0.507 | 0.343 |
| SRDCFdecon [30] | 0.648 | 0.483 | 0.630 | 0.447 | 0.443 | 0.327 |
| Staple [32] | 0.667 | 0.497 | 0.666 | 0.450 | 0.485 | 0.346 |
| ASLA [33] | 0.504 | 0.370 | 0.572 | 0.402 | 0.355 | 0.254 |

algorithm's performance under different conditions. In this section, the experimental setup is first described. Subsequently, detailed information about the proposed tracking algorithm and comparisons with several state-of-the-art tracking algorithms on each dataset are provided. Finally, qualitative evaluations are conducted on selected representative video sequences to further illustrate the strengths and weaknesses of the proposed algorithm.

## 4.1. Implementation details

Before correcting the errors caused by similar distractors, it is crucial to evaluate the degree of interference on the tracking results. This evaluation is performed using different thresholds across various datasets. In the TC-128 [28] dataset, the conditions for determining that the target is strongly interfered are $0 < \text{FSR} < 1.01$ and $\text{PSR} < 2.9$, and the maximum value area of the response map with cosine window, the maximum value area of the response map without cosine window and the second largest value area do not intersect. In the UAV123 [29] data set, the condition to determine that the target is strongly interfered is $0 < \text{FSR} < 1.02$ and $\text{PSR} < 2.8$, and the maximum value area of the response map with cosine window, the maximum value area of the response map without cosine window, and the second largest value area of the three areas do not intersect. In the UAV20L [29] data set, the conditions for determining that the target is strongly disturbed are $0 < \text{FSR} < 1.018$ and $\text{PSR} < 2.8$, and the maximum value area of the response map with cosine window, the maximum value area of the response map without cosine window and the second largest value area do not intersect. In TC-128 dataset and UAV123 dataset, when $V^t < 0$, the determination target changed drastically so that the samples were unreliable. In the UAV20L dataset, when $V^t < -0.1$ the determination target changed so drastically that the sample was unreliable.

## 4.2. Quantitative evaluation of our proposed tracking algorithm performance

### 4.2.1. Performance of our proposed tracking algorithm on TC-128 dataset. 
The fully convolutional Siamese network for temporal data association (SiamFC_TDA), which is proposed by us, is evaluated against five representative trackers in the TC-128 dataset [28]: SiamFC [14], Spatially Regularized Discriminative Correlation Filter with Deconvolution (SRDCFdecon) [32], Spatially Regularized Discriminative Correlation Filter (SRDCF) [31], Staple [32], and Adaptive Structured Local Analysis (ASLA) [33]. SiamFC serves as the baseline tracker for our proposed algorithm. Fig 4 presents the precision and success rate plots for these algorithms on the TC-128 dataset. Compared to the other trackers, SiamFC_TDA demonstrates superior performance in both the precision and success rate plots. Specifically, in the precision curve, SiamFC_TDA outperforms the second best performing SiamFC by 8.7%. In the success rate plot, SiamFC_TDA surpasses Staple, the second best performer, by 5% and improves upon the base tracker SiamFC by 6.6%.

To conduct a detailed analysis and comparison of the performance of each tracker, we evaluated their performance across 11 distinct attributes. The comparative performance of the seven tracking algorithms on these attributes is illustrated in Figs 5 and 6, and summarized in Table 2. Performance is measured in terms of both accuracy and success rate, and SiamFC_TDA outperforms all other algorithms in both metrics on seven attributes. Specifically, on the attribute of Motion Blur (MB), SiamFC_TDA exhibits the highest success rate. Across most attributes, SiamFC_TDA significantly improves upon the base tracker SiamFC, particularly in areas where SiamFC performs poorly, such as Background Clutter (BC), Deformation (DEF), In-Plane Rotation (IPR), Occlusion (OCC), Out-of-Plane Rotation (OPR), and Scale Change (SV).

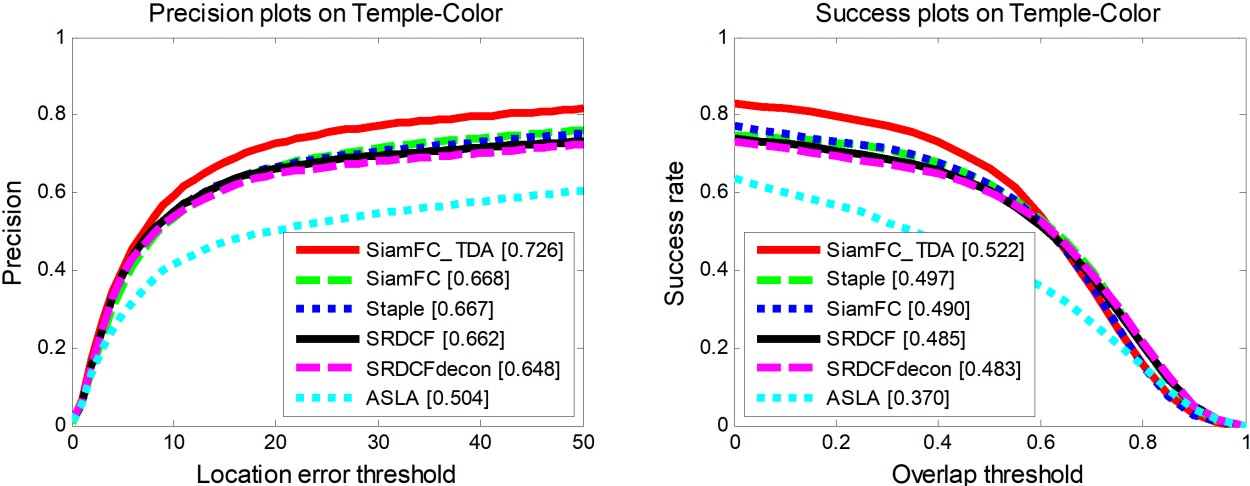

**Fig 4. Performance of SiamFC_TDA algorithm and comparison algorithm on TC-128.**

Since SiamFC relies solely on the first frame as a template and lacks the ability to adapt to appearance changes through template updates, it performs poorly in scenarios involving DEF, IPR, OCC, OPR, and SV. Conversely, our proposed reliable template update strategy, based on a reliable sample selection mechanism, not only adapts to appearance changes through template updates but also avoids template contamination during the update process in the presence of occlusion. Furthermore, the use of multi-template fusion allows the algorithm to effectively handle appearance changes, making it highly effective for dealing with changes in the target's appearance.

The notable improvement in BC attribute indicates that the proposed method of correcting the errors of similar distractors plays a crucial role in scenes with increasing interference enhancement, substantially improving the performance of the tracking algorithm. Different update processes generate template sets, which are then fused using the concept of ensemble learning. This approach further enhances the proposed tracking algorithm SiamFC_TDA by integrating the advantages of each template, enabling it to perform well in multiple challenging scenarios and adapt to a wider range of complex and changing scenes.

The extensive evaluation on the TC-128 dataset highlights the superior performance of SiamFC_TDA in terms of both accuracy and success rate. The proposed innovations, including a reliable template update strategy, error correction methods for similar distractors, and multi-template fusion, contribute significantly to the algorithm's robustness and reliability, positioning SiamFC_TDA as a leading solution for video tracking tasks.

**4.2.2. Performance of SiamFC_TDA on UAV123 dataset.** Six tracking algorithms were compared on the UAV123 dataset [27]: SiamFC_TDA, SiamFC [14], SRDCFdecon [28], SRDCF [29], Staple [30], and ASLA [31]. Fig 7 presents the accuracy and success rate curves for these tracking algorithms on the UAV123 dataset, which comprises aerial video sequences. The proposed tracking algorithm SiamFC_TDA demonstrates superior performance in both the accuracy and success rate curves. Specifically, in the precision curve plot, SiamFC_TDA achieves an average distance precision (DP) value of 72.1% with a threshold of 20 pixels, which is 3.6% higher than the second-best performing algorithm, SiamFC, the baseline tracker for SiamFC_TDA. In the success rate curve, SiamFC_TDA attains an area under curve

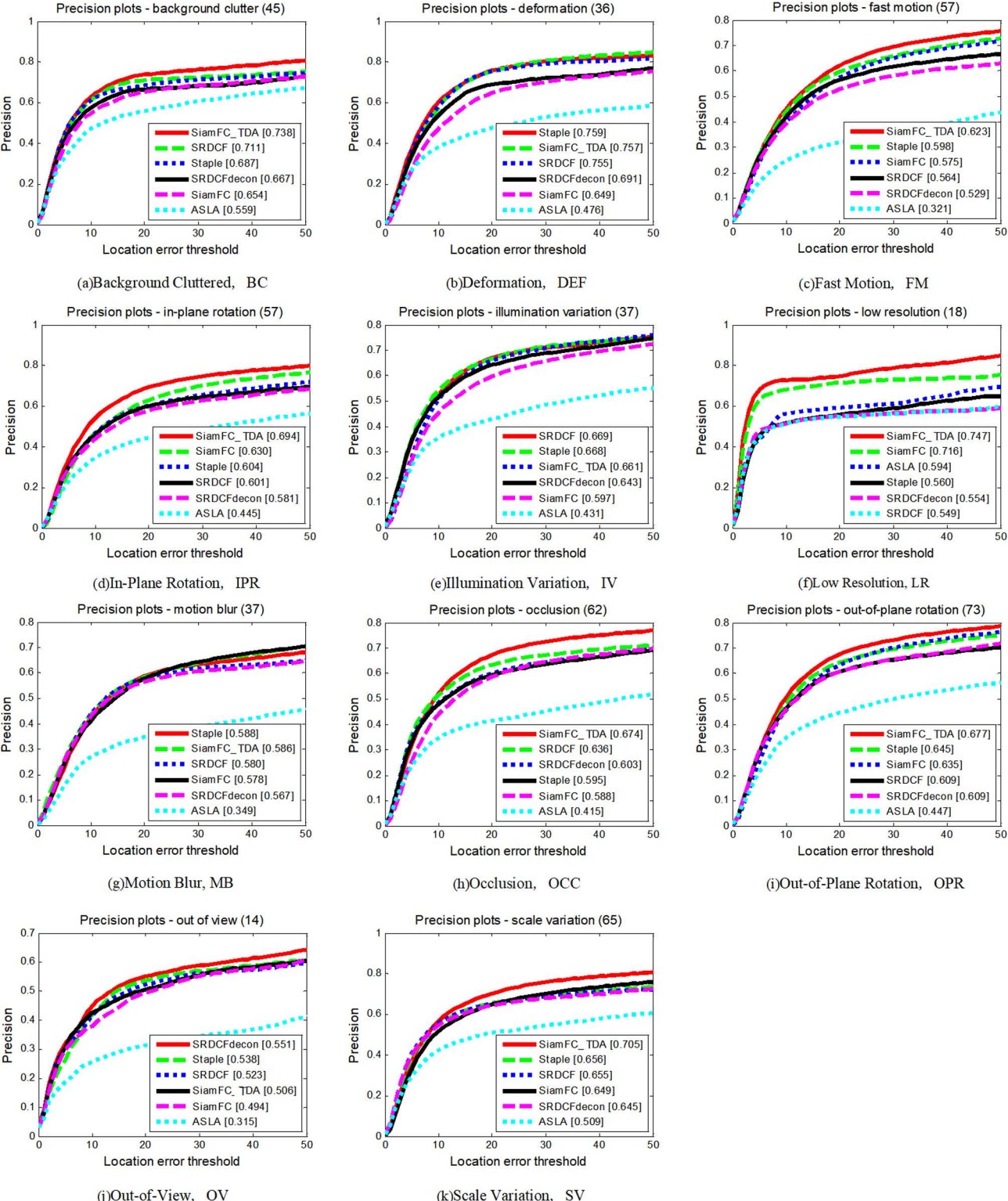

**Fig 5. Precision plots of SiamFC_TDA algorithm and comparison algorithm on TC-128 for each challenge attribute.**

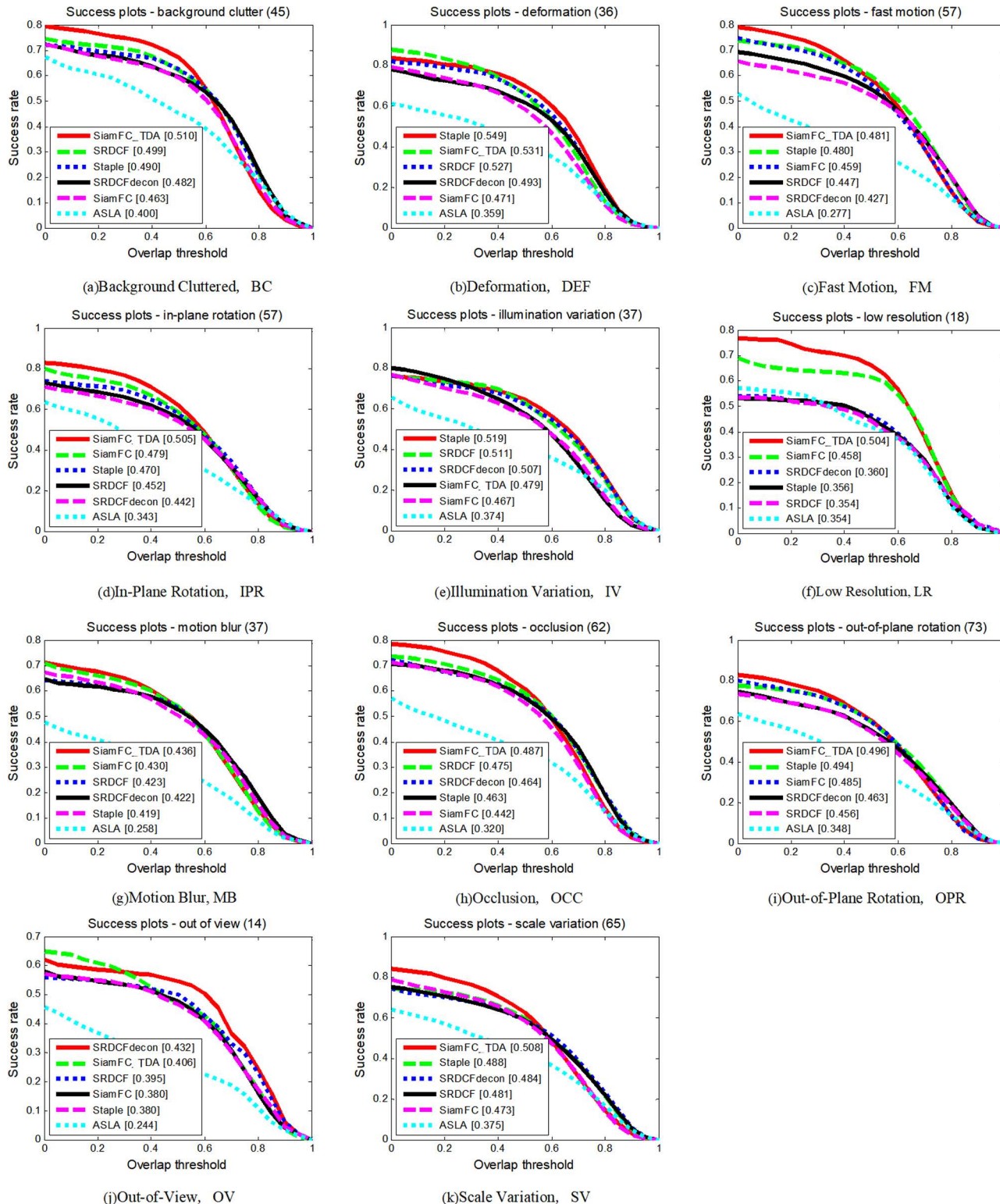

**Fig 6. Success plots of SiamFC_TDA algorithm and comparison algorithm on TC-128 for each challenge attribute.**

**Table 2. AUC values of SiamFC_TDA algorithm and comparison algorithm on each challenge attribute of TC_128. The higher the metrics, the better.**

|  | BC | DEF | FM | IPR | IV | LR | MB | OCC | OPR | OV | SV |
|---|---|---|---|---|---|---|---|---|---|---|---|
| SiamFC_TDA | **0.510** | 0.531 | **0.481** | **0.505** | 0.479 | **0.504** | **0.436** | **0.487** | **0.496** | 0.406 | **0.508** |
| SiamFC [14] | 0.463 | 0.471 | 0.459 | 0.479 | 0.467 | 0.458 | 0.430 | 0.442 | 0.485 | 0.380 | 0.473 |
| SRDCF [31] | 0.499 | 0.527 | 0.447 | 0.452 | 0.511 | 0.354 | 0.423 | 0.475 | 0.456 | 0.395 | 0.481 |
| SRDCFdecon [30] | 0.482 | 0.493 | 0.427 | 0.442 | 0.507 | 0.360 | 0.422 | 0.464 | 0.463 | **0.432** | 0.484 |
| Staple [32] | 0.490 | **0.549** | 0.480 | 0.470 | **0.519** | 0.356 | 0.419 | 0.463 | 0.494 | 0.380 | 0.488 |
| ASLA [33] | 0.400 | 0.359 | 0.277 | 0.343 | 0.374 | 0.354 | 0.258 | 0.320 | 0.348 | 0.244 | 0.375 |

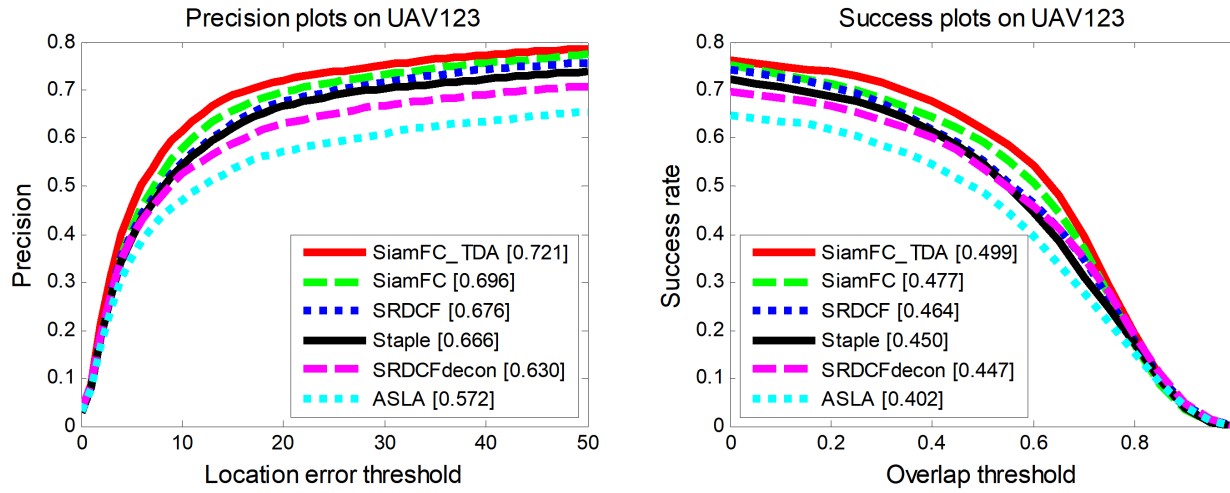

**Fig 7. Performance of SiamFC_TDA algorithm and comparison algorithm on UAV123.**

(AUC) value of 49.9%, which is 4.6% higher than the second-best performing baseline tracker SiamFC.

The UAV123 dataset introduces several unique challenges compared to the TC-128 dataset. Video sequences in the UAV123 dataset are annotated with 12 video attribute labels. The comparative performance of the seven tracking algorithms across these 12 attributes is depicted in Figs 8 and 9, and summarized in Table 3. Performance is measured in terms of both accuracy and success rate, and SiamFC_TDA excels in both metrics on nine attributes. Notably, on the attribute of BC, SiamFC_TDA exhibits the highest accuracy. Across most attributes, SiamFC_ TDA shows improved performance compared to the baseline tracker SiamFC.

These enhancements are primarily attributed to the proposed reliable template update strategy based on a reliable sample selection mechanism, the method to correct the errors caused by similar distractors, and the application of multi-template fusion. These innovations are particularly effective in addressing the most challenging scenarios, such as those involving BC, FM, and IV. The robust template update strategy ensures that the tracking model adapts dynamically to changes in the target appearance, thereby maintaining high accuracy even in complex environments. The multi-template fusion technique leverages multiple templates to enhance the system's resilience against occlusions and other visual disturbances, further improving the overall tracking performance.

The UAV123 dataset's diverse set of video attributes provides a comprehensive evaluation framework for assessing the robustness of tracking algorithms. SiamFC_TDA's superior

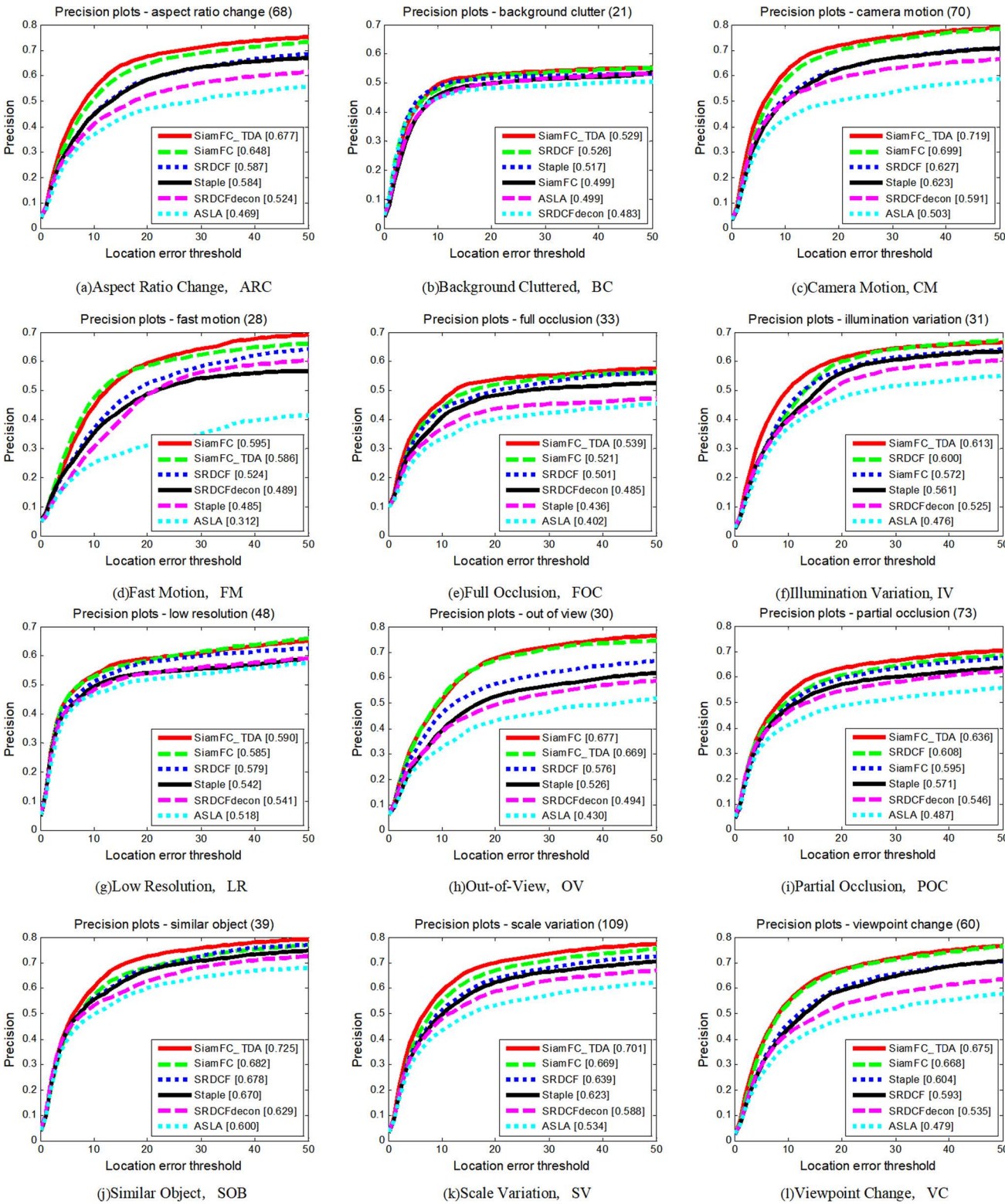

**Fig 8. Precision plots of SiamFC_TDA algorithm and comparison algorithm on UAV123 for each challenge attribute.**

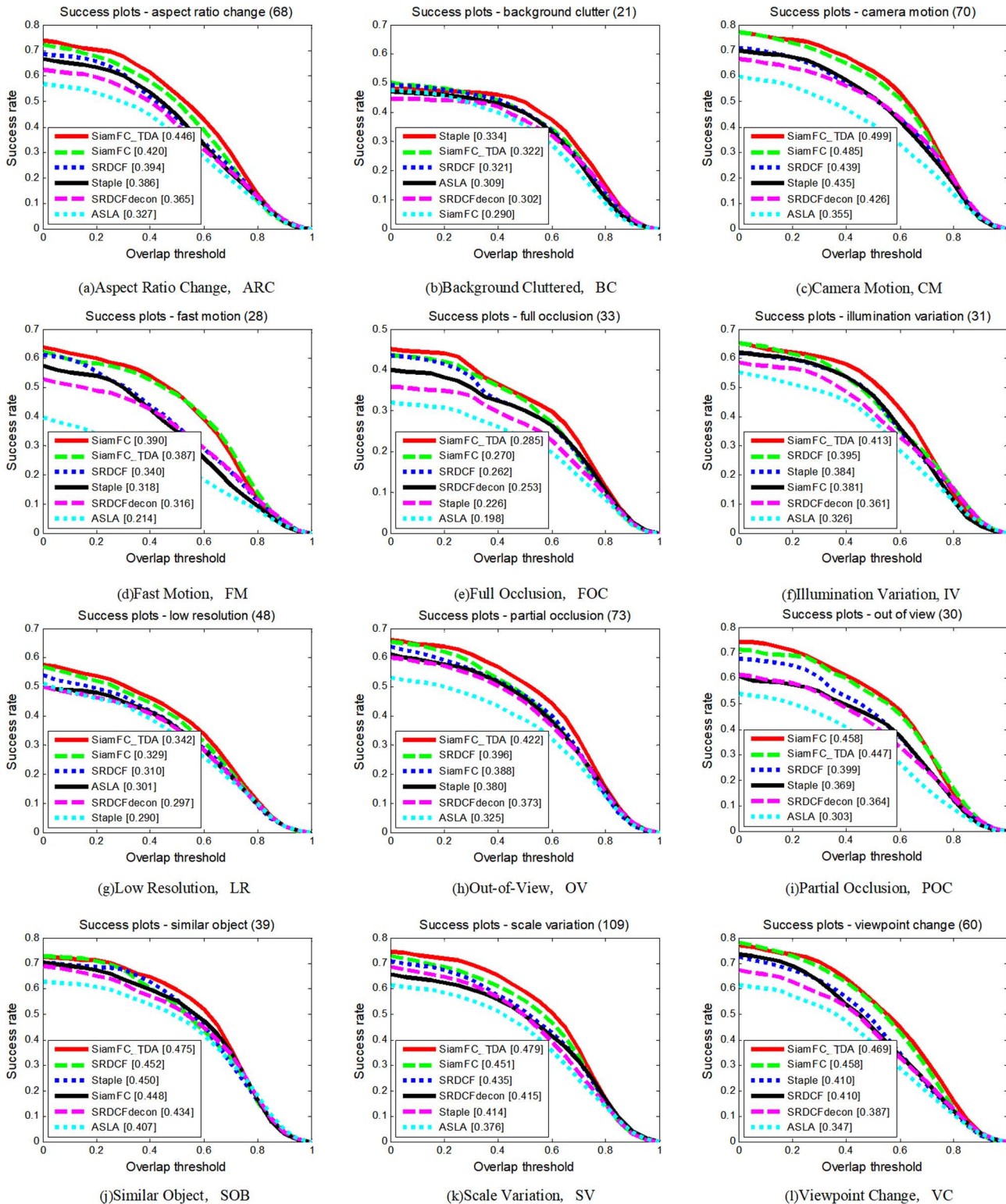

**Fig 9. Success plots of SiamFC_TDA algorithm and comparison algorithm on UAV123 for each challenge attribute.**

**Table 3. AUC values of SiamFC_TDA algorithm and comparison algorithm on each challenge attribute of UAV123. The higher the metrics, the better.**

|  | ARC | BC | CM | FM | FO | IV | LR | OV | PO | SO | SV | VC |
|---|---|---|---|---|---|---|---|---|---|---|---|---|
| SiamFC_TDA | **0.446** | 0.322 | **0.499** | 0.387 | **0.285** | **0.413** | **0.342** | 0.447 | **0.422** | **0.517** | **0.479** | **0.469** |
| SiamFC [14] | 0.420 | 0.290 | 0.485 | **0.390** | 0.270 | 0.395 | 0.329 | **0.458** | 0.388 | 0.512 | 0.451 | 0.458 |
| SRDCF [31] | 0.394 | 0.321 | 0.439 | 0.340 | 0.262 | 0.395 | 0.310 | 0.399 | 0.396 | 0.452 | 0.435 | 0.410 |
| SRDCFdecon [30] | 0.365 | 0.302 | 0.426 | 0.316 | 0.253 | 0.361 | 0.297 | 0.364 | 0.373 | 0.434 | 0.415 | 0.387 |
| Staple [32] | 0.386 | **0.334** | 0.435 | 0.318 | 0.226 | 0.384 | 0.290 | 0.369 | 0.380 | 0.450 | 0.414 | 0.410 |
| ASLA [33] | 0.327 | 0.309 | 0.355 | 0.214 | 0.198 | 0.326 | 0.301 | 0.303 | 0.325 | 0.407 | 0.376 | 0.347 |

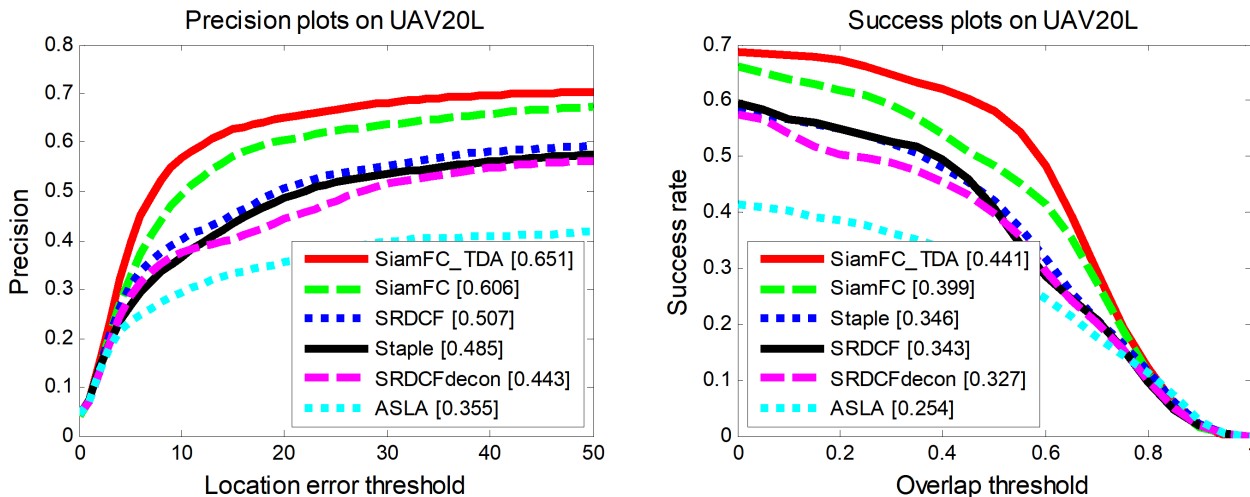

**Fig 10. Performance of SiamFC_TDA algorithm and comparison algorithm on UAV20L.**

performance across these attributes underscores its capability to maintain stable tracking even in challenging and dynamic aerial video sequences. This robustness is critical for real-world applications where consistent and reliable tracking is paramount.

**4.2.3. Performance of SiamFC_TDA on UAV20L dataset.** The proposed tracking algorithm SiamFC_TDA has been rigorously evaluated against six other tracking algorithms on the challenging UAV20L dataset [29]: SiamFC [14], SRDCFdecon [30], SRDCF [31], Staple [32], and ASLA [33]. The UAV20L dataset is specifically designed for long aerial video sequences, presenting a more significant challenge than the UAV123 dataset and requiring higher robustness from the tracking algorithms.

As illustrated in Fig 10, SiamFC_TDA demonstrates superior performance in both accuracy and success rate metrics. In the precision curve plot, the proposed tracking algorithm SiamFC_TDA achieves an average DP value of 65.1% with a threshold of 20 pixels. This performance is 7.5% higher than the second-best performing algorithm, SiamFC, which serves as the baseline tracker for SiamFC_TDA. Furthermore, in the success rate curve, SiamFC_TDA attains an AUC value of 44.1%, surpassing the second-best performer, ECO, by 1.4% and outperforming the baseline tracker SiamFC by 10.6%.

As depicted in Figs 11 and 12, and summarized in Table 4, SiamFC_TDA outperforms all other algorithms across various attributes, with the exception of Low Resolution (LR) in terms of precision rate and Similar Objects (SOB) in terms of AUC values. This exceptional performance underscores the robustness of the proposed algorithm, even in highly complex and

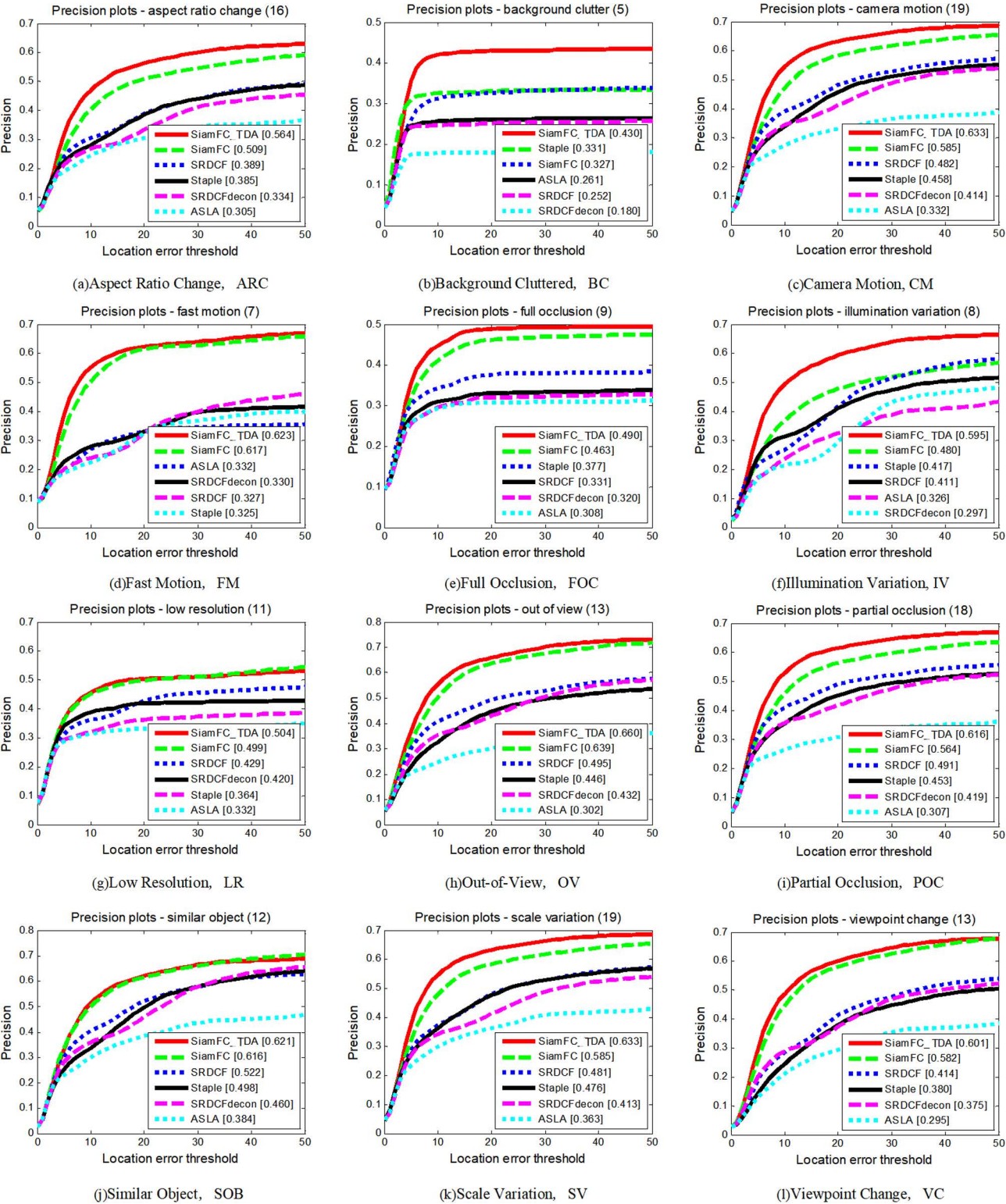

**Fig 11. Precision plots of SiamFC_TDA algorithm and comparison algorithm on UAV20L for each challenge attribute.**

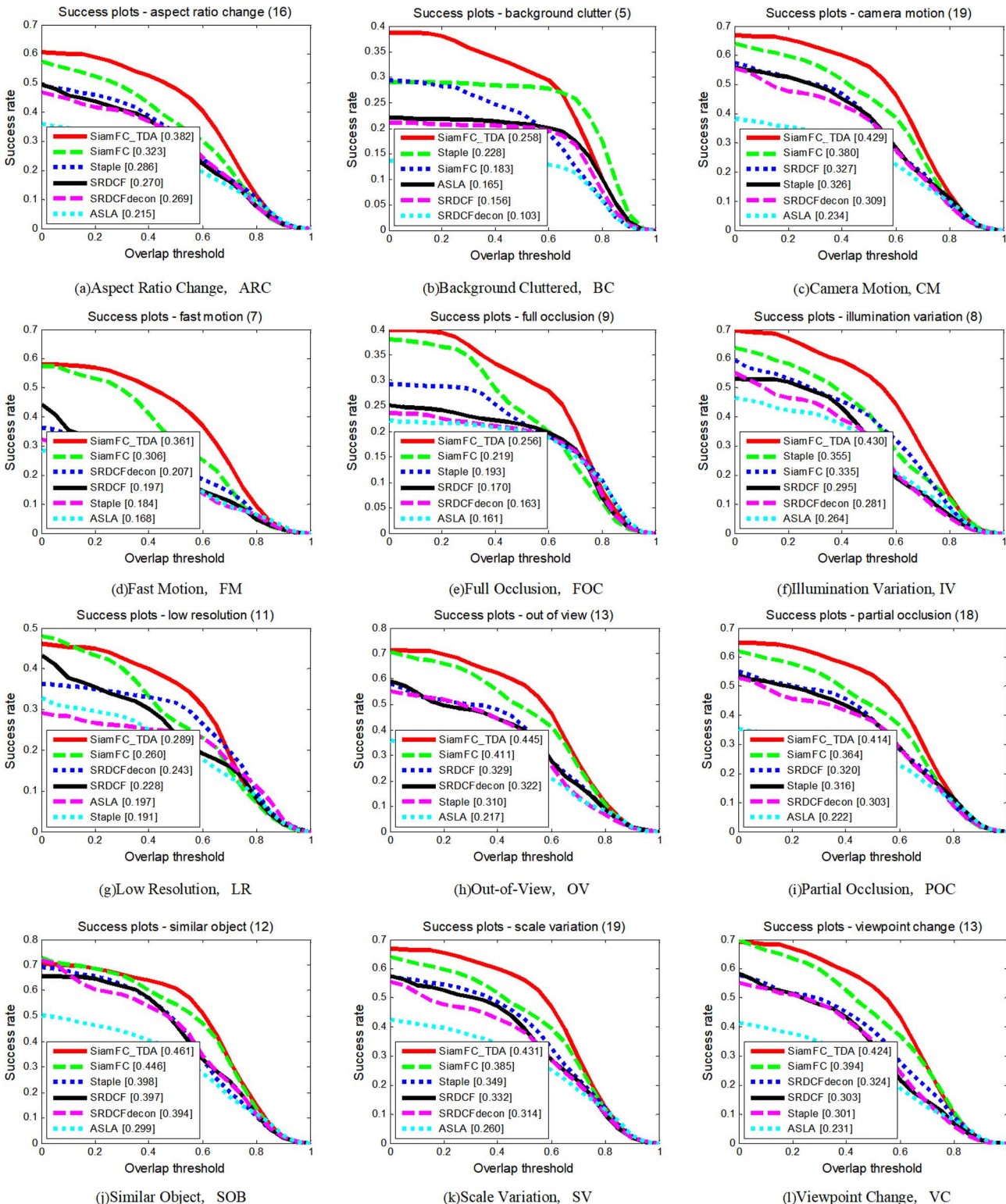

**Fig 12. Success plots of SiamFC_TDA algorithm and comparison algorithm on UAV20L for each challenge attribute.**

**Table 4. AUC values of SiamFC_TDA algorithm and comparison algorithm on each challenge attribute of UAV20L. The higher the metrics, the better.**

|  | ARC | BC | CM | FM | FO | IV | LR | OV | PO | SO | SV | VC |
|---|---|---|---|---|---|---|---|---|---|---|---|---|
| SiamFC_TDA | **0.382** | **0.258** | **0.429** | **0.361** | **0.256** | **0.430** | **0.289** | **0.445** | **0.414** | **0.461** | **0.431** | **0.424** |
| SiamFC [14] | 0.323 | 0.183 | 0.380 | 0.306 | 0.219 | 0.335 | 0.260 | 0.411 | 0.364 | 0.446 | 0.385 | 0.394 |
| SRDCF [31] | 0.270 | 0.156 | 0.327 | 0.197 | 0.170 | 0.295 | 0.228 | 0.329 | 0.320 | 0.397 | 0.332 | 0.303 |
| SRDCFdecon [30] | 0.269 | 0.103 | 0.309 | 0.207 | 0.163 | 0.281 | 0.243 | 0.322 | 0.303 | 0.394 | 0.314 | 0.324 |
| Staple [32] | 0.286 | 0.228 | 0.326 | 0.184 | 0.193 | 0.355 | 0.191 | 0.310 | 0.316 | 0.398 | 0.349 | 0.301 |
| ASLA [33] | 0.215 | 0.165 | 0.234 | 0.168 | 0.161 | 0.264 | 0.197 | 0.217 | 0.222 | 0.299 | 0.260 | 0.231 |

challenging long video sequences. The robustness of SiamFC_TDA is attributed to its effective solutions for addressing common issues such as sample contamination, template contamination, and interference.

The UAV20L dataset presents several unique challenges that exacerbate the difficulties faced by tracking algorithms. Long aerial video sequences often contain dynamic scenes with significant variations in lighting, weather conditions, and camera movements, which can lead to frequent occlusions and rapid changes in target appearance. Despite these challenges, SiamFC_TDA consistently maintains high tracking accuracy and stability, demonstrating its ability to handle complex scenarios effectively.

The improvements introduced by SiamFC_TDA, such as enhanced feature extraction techniques and adaptive learning mechanisms, contribute significantly to its superior performance. These enhancements enable the algorithm to adapt quickly to changing conditions and maintain consistent tracking accuracy, even in the presence of occlusions and other visual disturbances. The robustness of SiamFC_TDA is further evidenced by its consistent performance across various attributes, indicating its suitability for a wide range of applications in aerial surveillance and tracking.

The comprehensive evaluation on the UAV20L dataset highlights the robustness and effectiveness of the proposed SiamFC_TDA tracking algorithm. Its superior performance in terms of both precision and success rates, along with its ability to maintain stable tracking under challenging conditions, positions SiamFC_TDA as a leading solution for long aerial video sequence tracking tasks.

### 4.3. Qualitative evaluation of SiamFC_TDA algorithm performance

Fig 13 illustrates the qualitative outcomes of six different tracking algorithms, including SiamFC_TDA, SiamFC [14], SRDCFdecon [30], SRDCF [31], Staple [32], and ASLA [33], evaluated across three distinct video sequences. These sequences—Busstation_ce2, Bicycle, and Basketball_ce3—are characterized by their inclusion of multiple challenge attributes, such as occlusion, lighting, and cluttered background, which reflect the complexity and variability inherent in real-world tracking scenarios. As shown in Fig 13, the primary difficulty arises from occlusion, which commonly occurs when individuals overlap one another or are obscured by other objects. This phenomenon poses significant challenges for tracking algorithms, leading to the potential misidentification of targets. Occlusion not only complicates the tracking process but also disrupts the continuity of time series data, making it difficult for algorithms to maintain consistent and accurate target identification throughout the sequence. The results presented in Fig 13 highlight the superior performance of SiamFC_TDA compared to the other algorithms, particularly when contrasted with the baseline SiamFC approach. Consequently, the qualitative assessment underscores the robustness and effectiveness of the proposed SiamFC_TDA algorithm in handling intricate environments.

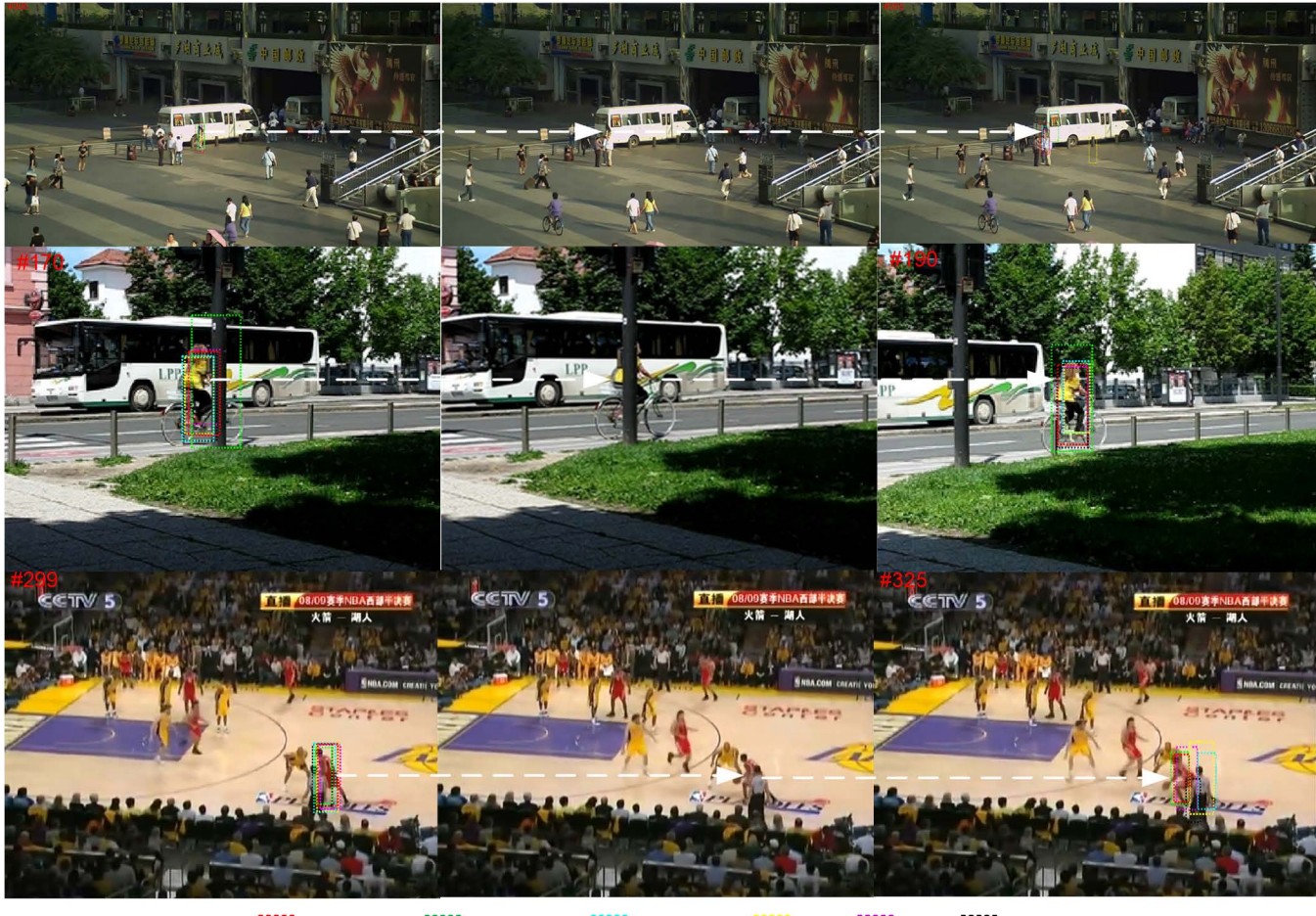

**Fig 13. Visualization of tracking results of SiamFC_TDA algorithm and comparison algorithm on challenging sequences.**

## 5. Conclusion

In this paper, we propose a novel approach for multi-template tracking to enhance the coherence and consistency of time series data in complex scenes. Our method addresses the limitations of the original Siamese network tracking algorithm, which often struggles with significant variations in target appearance and environmental conditions such as occlusions and background clutter. To achieve this, we introduce a multi-template fusion strategy that combines multiple online update templates with the initial frame template. This combination allows us to leverage the adaptability of the online update templates to cope with changes in the target's appearance, while the initial frame template helps mitigate the effects of occlusions and background noise. By fusing these templates, our approach corrects for potential template contamination in challenging environments. Furthermore, we propose two key strategies to improve the reliability of the tracking process: Interference-aware strategy minimizes the generation of incorrect samples by filtering out unreliable data points, ensuring that only high-quality samples contribute to the tracking process. Sample reliability-aware strategy blocks the propagation of erroneous information to the templates, thereby preventing the

degradation of tracking performance due to contaminated data. Experimental evaluations conducted on extensive datasets demonstrate the effectiveness of our proposed algorithm. It not only enhances the robustness of the Siamese network tracking algorithm in complex scenes but also significantly improves its tracking accuracy.

## Supporting information

**S1 File. Results_TC-128.** The results of the experiment on TC-128.
(RAR)

**S2 File. Results_UAV123.** The results of the experiment on UAV123.
(RAR)

**S3 File. Results_UAV20L.** The results of the experiment on UAV20L.
(RAR)

## Author contributions

**Conceptualization:** Yi Tao.

**Data curation:** Fei Wang, Mohan Li.

**Formal analysis:** Yi Tao, Juncheng Zhou.

**Investigation:** Fei Wang.

**Methodology:** Fei Wang.

**Project administration:** Yi Tao.

**Resources:** Yi Tao.

**Software:** Fei Wang.

**Supervision:** Yi Tao.

**Validation:** Fei Wang.

**Visualization:** Fei Wang.

**Writing – original draft:** Fei Wang.

**Writing – review & editing:** Fei Wang, Jie Liu, Bo Dong, Ruidong Liu, Sihao Chen, Kan Jiao.

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
