## [Decision Letter · Decision Letter 0]

11 Dec 2024

PONE-D-24-52028Improved siamese tracking for temporal data associationPLOS ONE

Dear Dr. Wang,

Thank you for submitting your manuscript to PLOS ONE. After careful consideration, we feel that it has merit but does not fully meet PLOS ONE’s publication criteria as it currently stands. Therefore, we invite you to submit a revised version of the manuscript that addresses the points raised during the review process.

We look forward to receiving your revised manuscript.

Kind regards,

Xin Xu, Ph.D.

Academic Editor

PLOS ONE

Journal Requirements:

Please confirm at this time whether or not your submission contains all raw data required to replicate the results of your study. Authors must share the “minimal data set” for their submission. PLOS defines the minimal data set to consist of the data required to replicate all study findings reported in the article, as well as related metadata and methods (https://journals.plos.org/plosone/s/data-availability#loc-minimal-data-set-definition ).

If your submission does not contain these data, please either upload them as Supporting Information files or deposit them to a stable, public repository and provide us with the relevant URLs, DOIs, or accession numbers. For a list of recommended repositories, please see https://journals.plos.org/plosone/s/recommended-repositories .

6. Please include a separate caption for each figure in your manuscript.

Reviewers' comments:

Reviewer's Responses to Questions

**Comments to the Author**

1. Is the manuscript technically sound, and do the data support the conclusions?

Reviewer #1: Yes

Reviewer #2: Yes

2. Has the statistical analysis been performed appropriately and rigorously? 

Reviewer #1: Yes

Reviewer #2: Yes

3. Have the authors made all data underlying the findings in their manuscript fully available?

Reviewer #1: Yes

Reviewer #2: Yes

4. Is the manuscript presented in an intelligible fashion and written in standard English?

Reviewer #1: Yes

Reviewer #2: Yes

5. Review Comments to the Author

Reviewer #1: 1. All figures are missing, please update them.

2. In the section 2.3 Multi-branch tracking fusion, the relevance of related work and the shortcomings of existing work should be spelled out.

3. In Eq.4 - Eq. 7, the physical meaning of each variable needs to be elaborated.

4. In all tables, the best performance needs to be bolded for easy viewing

5. This paper needs to cite more relevant papers such as [1-3].

[1] Towards generalizable person re-identification with a bi-stream generative model. Pattern Recognition

[2] Searching parameterized retrieval & verification loss for re-identification. IEEE Journal of Selected Topics in Signal Processing

[3] Mix-Modality Person Re-Identification: A New and Practical Paradigm. arxiv

Reviewer #2: 1, The clarity of the article's diagrams is a concern and the diagrams are not sufficiently clear, e.g. the lack of a demonstration of the impact of occlusion in the introduction.

2, On page 3, line 61, how does the temporal data contain the size of the target?

3, Many of the formulas in the article lack explanation of the meaning of the characters.

4, The methods compared in the experimental section need to be given in the table with the source of the method and an indication as to whether the metrics are higher or lower, and the optimal performance needs to be given a special notation to make it clearer.

5, Please add citations for the following papers:

1. Low-Light Salient Object Detection Meets the Small Size. IEEE Transactions on Emerging Topics in Computational Intelligence

2. Informative Classes Matter: Towards Unsupervised Domain Adaptive Nighttime Semantic Segmentation. in Proceedings of the 31th ACM International Conference on Multimedia (ACM MM), 2023

3. Exploring Image Enhancement for Salient Object Detectionin Low Light Images. ACM Transactions on Multimedia Computing, Communications, and Applications. 2021

6. PLOS authors have the option to publish the peer review history of their article (what does this mean? ). If published, this will include your full peer review and any attached files.

**Do you want your identity to be public for this peer review?** For information about this choice, including consent withdrawal, please see our Privacy Policy .

Reviewer #1: No

Reviewer #2: No

---

## [Author Response · Author response to Decision Letter 0]

13 Feb 2025

Dear Editors and Reviewers:

Thank you for your letter and for the reviewers’ comments concerning our manuscript entitled “Improved siamese tracking for temporal data association” (ID: PONE-D-24-52028). Those comments are all valuable and very helpful for revising and improving our paper, as well as the important guiding significance to our researches. We have studied comments carefully and have made corrections which we hope meet with approval. The main corrections in the paper and the responds to the reviewer’s comments are as follows:

Reviewer1:

1.All figures are missing, please update them.

Reply: All figures will be updated and checked for proper formatting and resolution.

2.In the section 2.3 Multi-branch tracking fusion, the relevance of related work and the shortcomings of existing work should be spelled out.

Reply: Considering the Reviewer’s suggestion, in the section 2.3, “Thus, there are many fusions of multiple divergent trackers.” was changed to “In the pursuit of more precise and reliable object tracking, the concept of multi-branch tracking fusion holds significant relevance. As we've established, the ever-changing nature of target appearances demands adaptable tracking mechanisms. Multi-branch tracking fusion, with its various manifestations like Multi-domain convolutional neural networks (MDNet) [19], Tracking adaptation context-aware auto-encoders (TRACA) [20], Multi-branch Siamese tracking (MBST) [21], and Multi-features Siamese tracker (MFST) [22], attempts to counter the challenges posed by diverse target conditions.” to highlight the relevance of related work. And “However, a notable shortcoming is that the independence of the domains might lead to suboptimal integration during actual tracking.”, “While this context-based selection mechanism is innovative, it suffers from high computational complexity.”,“Although it provides a practical way to adapt to varying target appearances, it heavily depends on the quality and diversity of the predefined feature representations.”,“This fusion aims to enhance the tracking performance by combining different levels of semantic and visual information. Nevertheless, the challenge lies in determining the optimal combination weights for the different features. An improper weighting scheme can either overemphasize less relevant features or underutilize crucial ones, leading to subpar tracking accuracy.” were added to emphasize the shortcomings of existing work.

3.In Eq.4 - Eq. 7, the physical meaning of each variable needs to be elaborated.

Reply: “Where and are the corresponding weighting coefficients in the X-axis and Y-axis directions, which are obtained by trigonometric functions according to the approximate angle of the region. ” were added after Eq.4. The physical meanings of , , and were elaborated.

4.In all tables, the best performance needs to be bolded for easy viewing.

Reply: The best performance metrics in all tables have been bolded for clarity.

5.This paper needs to cite more relevant papers such as [1-3].

[1]Towards generalizable person re-identification with a bi-stream generative model. Pattern Recognition

[2] Searching parameterized retrieval & verification loss for re-identification. IEEE Journal of Selected Topics in Signal Processing

[3] Mix-Modality Person Re-Identification: A New and Practical Paradigm. Arxiv

Reply: Papers [1-3] have been cited in relevant sections.

Reviewer2:

1.The clarity of the article's diagrams is a concern and the diagrams are not sufficiently clear, e.g. the lack of a demonstration of the impact of occlusion in the introduction.

Reply: Figures were re-generated with higher resolution. Fig 12 was changed to illustrate occlusion challenges. In the introduction, a figure was added to demonstrate the impact of occlusion. And the content describing the impact of occlusion is changed to “As illustrated in Fig 1, occlusions, where objects overlap or are partially hidden by other elements within the scene, are particularly problematic. These occlusions can cause tracking algorithms to lose sight of the target, leading to incorrect associations and disrupted temporal coherence, and complicate post-occlusion recovery due to appearance changes or similar objects.”

2.On page 3, line 61, how does the temporal data contain the size of the target?

Reply: Temporal data contains the target's size by recording the dimensions (e.g., width and height) of the bounding box enclosing the object in each frame. These dimensions are aggregated across consecutive frames, forming a size trajectory that reflects how the target's size evolves over time as part of its dynamic spatial properties.

3.Many of the formulas in the article lack explanation of the meaning of the characters.

Reply: “Where and are the updated parameters and is a learning rate.” was added after Eq.1. “Where and are the corresponding weighting coefficients in the X-axis and Y-axis directions, which are obtained by trigonometric functions according to the approximate angle of the region. ” was added after Eq.4. The physical meanings of , , and were elaborated.

4.The methods compared in the experimental section need to be given in the table with the source of the method and an indication as to whether the metrics are higher or lower, and the optimal performance needs to be given a special notation to make it clearer.

Reply: Comparison methods included references in the table. The best performance metrics in all tables have been bolded for clarity. “The higher the metrics, the better.” was added to the title of each table.

5.Please add citations for the following papers:

1. Low-Light Salient Object Detection Meets the Small Size. IEEE Transactions on Emerging Topics in Computational Intelligence

2. Informative Classes Matter: Towards Unsupervised Domain Adaptive Nighttime Semantic Segmentation. in Proceedings of the 31th ACM International Conference on Multimedia (ACM MM), 2023

3. Exploring Image Enhancement for Salient Object Detectionin Low Light Images. ACM Transactions on Multimedia Computing, Communications, and Applications. 2021

Reply: Papers [1-3] have been cited in relevant sections.

Special thanks to you for your good comments.

We tried our best to improve the manuscript and made some changes in the manuscript. These changes will not influence the content and framework of the paper.

We appreciate for Editors’ and Reviewers’ warm work earnestly, and hope that the correction will meet with approval.

Once again, thank you very much for your comments and suggestions.

Yours sincerely,

Dr Fei Wang

---

## [Decision Letter · Decision Letter 1]

25 Feb 2025

Improved siamese tracking for temporal data association

PONE-D-24-52028R1

Dear Dr. Wang,

We’re pleased to inform you that your manuscript has been judged scientifically suitable for publication and will be formally accepted for publication once it meets all outstanding technical requirements.

Kind regards,

Xin Xu, Ph.D.

Academic Editor

PLOS ONE

Additional Editor Comments (optional):

Reviewers' comments:

Reviewer's Responses to Questions

**Comments to the Author**

1. If the authors have adequately addressed your comments raised in a previous round of review and you feel that this manuscript is now acceptable for publication, you may indicate that here to bypass the “Comments to the Author” section, enter your conflict of interest statement in the “Confidential to Editor” section, and submit your "Accept" recommendation.

Reviewer #1: All comments have been addressed

Reviewer #2: All comments have been addressed

2. Is the manuscript technically sound, and do the data support the conclusions?

Reviewer #1: Yes

Reviewer #2: Yes

3. Has the statistical analysis been performed appropriately and rigorously? 

Reviewer #1: Yes

Reviewer #2: Yes

4. Have the authors made all data underlying the findings in their manuscript fully available?

Reviewer #1: Yes

Reviewer #2: Yes

5. Is the manuscript presented in an intelligible fashion and written in standard English?

Reviewer #1: Yes

Reviewer #2: Yes

6. Review Comments to the Author

Reviewer #1: Thanks for authors. All comments have been addressed. Looking forward to the publication of the paper.

Reviewer #2: (No Response)

7. PLOS authors have the option to publish the peer review history of their article (what does this mean? ). If published, this will include your full peer review and any attached files.

**Do you want your identity to be public for this peer review?** For information about this choice, including consent withdrawal, please see our Privacy Policy .

Reviewer #1: No

Reviewer #2: No

---

## [Editor Report · Acceptance letter]

PONE-D-24-52028R1

PLOS ONE

Dear Dr. Wang,

I'm pleased to inform you that your manuscript has been deemed suitable for publication in PLOS ONE. Congratulations! Your manuscript is now being handed over to our production team.

Kind regards,

on behalf of

Dr. Xin Xu

Academic Editor

PLOS ONE